# Comparing Bayesian spatial models: Goodness-of-smoothing criteria for assessing under- and over-smoothing

**Earl W. Duncan** *, **Kerrie L. Mengersen**

ARC Centre of Excellence for Mathematical and Statistical Frontiers, Queensland University of Technology, Brisbane, Australia

* earl.w.duncan@gmail.com

## Abstract

### Background

Many methods of spatial smoothing have been developed, for both point data as well as areal data. In Bayesian spatial models, this is achieved by purposefully designed prior(s) or smoothing functions which smooth estimates towards a local or global mean. Smoothing is important for several reasons, not least of all because it increases predictive robustness and reduces uncertainty of the estimates. Despite the benefits of smoothing, this attribute is all but ignored when it comes to model selection. Traditional goodness-of-fit measures focus on model fit and model parsimony, but neglect "goodness-of-smoothing", and are therefore not necessarily good indicators of model performance. Comparing spatial models while taking into account the degree of spatial smoothing is not straightforward because smoothing and model fit can be viewed as opposing goals. Over- and under-smoothing of spatial data are genuine concerns, but have received very little attention in the literature.

### Methods

This paper demonstrates the problem with spatial model selection based solely on goodness-of-fit by proposing several methods for quantifying the degree of smoothing. Several commonly used spatial models are fit to real data, and subsequently compared using the goodness-of-fit and goodness-of-smoothing statistics.

### Results

The proposed goodness-of-smoothing statistics show substantial agreement in the task of model selection, and tend to avoid models that over- or under-smooth. Conversely, the traditional goodness-of-fit criteria often don't agree, and can lead to poor model choice. In particular, the well-known deviance information criterion tended to select under-smoothed models.

### Conclusions

Some of the goodness-of-smoothing methods may be improved with modifications and better guidelines for their interpretation. However, these proposed goodness-of-smoothing

**Data Availability Statement:** All data are publicly available from existing studies, the details of which are provided in the paper.

**Funding:** This work was supported by the ARC Centre of Excellence for Mathematical and

Statistical Frontiers (ACEMS) and Queensland
University of Technology (QUT).

**Competing interests:** The authors have declared
that no competing interests exist.

**Abbreviations:** AIC, Akaike's information criterion;
BIC, Bayesian information criterion; BYM, a
Bayesian spatial model named after the authors
Besag, York, and Mollié; CARSIR, covariate-
adjusted raw standardised incidence ratio; CASIR,
covariate-adjusted standardised incidence ratio;
CPO, conditional predictive ordinate; DIC, deviance
information criterion; GoF, goodness-of-fit; GoS,
goodness-of-smoothing; ICAR, intrinsic conditional
autoregressive; IG, inverse-gamma; LTN, left-
truncated Normal; MCMC, Markov chain Monte
Carlo; PPC, posterior predictive check; PPO,
posterior predictive ordinate; SIDS, sudden infant
death syndrome; SIR, standardised incidence ratio;
SRE, spatial random effect; SSRE, structured
spatial random effect; USRE, unstructured spatial
random effect; WAIC, Widely applicable
information criterion.

methods offer researchers a solution to spatial model selection which is easy to implement.
Moreover, they highlight the danger in relying on goodness-of-fit measures when comparing
spatial models.

## Introduction

Spatial smoothing is a technique used when modelling the underlying data-generating process
of spatial data to account for spatial autocorrelation, as expressed by Tobler's first law of geog-
raphy: ". . . near things are more related than distant things" [1]. Neglecting spatial autocorre-
lation is akin to ignoring the order of time-series data, leading to greater uncertainty about the
model parameters, poorer predictions, and misguided inference. Conversely, when spatial
smoothing is applied, it has the benefit of more appropriately representing the statistical uncer-
tainty of model parameters, better predictions, and providing more insight into the layers of
the underlying data-generating process, similar to how the trend and seasonality help explain
layers of decomposed time series data [2, 3].

Many methods of spatial smoothing have been developed, for both point data as well as
areal data, including linear or non-linear functions based on distances, loess smoothing [4],
spline functions [5], kriging [6] and Gaussian process priors [7, 8], and empirical Bayes
approaches to spatial smoothing [9]. For an overview of smoothing techniques, see Kafadar [2]
and Tiwari and Rushton [10]. Empirical Bayes methods, which smooth estimates of points on
a spatial surface towards the global mean based on a distribution whose parameters are fixed *a
priori*, gained popularity as computing power increased and parameter estimation techniques
became more widely accessible. Fully Bayes methods have also been proposed [10, 11], where
the surface is typically estimated by one or more spatially varying parameters with purposefully
chosen prior distributions to account for the spatial autocorrelation.

Spatial smoothing plays an important role in a broad range of applications, including the
assessment of feature significance [12] and seafloor classification in geostatistics [13], monitor-
ing of groundwater contaminant plumes [14], image processing [11, 15], the calorific value
distributions in coal facies [16], and analysis of traffic accidents [17, 18] to name a few.

Notwithstanding the benefits associated with spatial smoothing, there seems to be a small
but growing awareness of the dangers associated with under- and over-smoothing. While
over-smoothing causes genuine deviations from the local or global mean to be obscured [13,
19], under-smoothing is equally undesirable as it exaggerates features in the surface, making
them indistinguishable from background noise, which defeats the point of spatial smoothing.
The negative effects of under-smoothing are a lot less vocalised in spatial modelling than in
time series modelling, where the link between under-smoothing and residual autocorrelation,
large prediction errors, and biased hypothesis tests have been articulated [20, 21].

Despite the growing awareness, there is very little guidance in the literature on how to assess
the appropriateness of the level of spatial smoothing, and according to our knowledge, any
efforts to account for such smoothing in model selection are non-existent. The latter is evident
by the widespread use of model selection criteria like the Bayesian information criterion (BIC)
[22], the deviance and related deviance information criterion (DIC) [23], and widely applica-
ble information criterion (WAIC) [24] to compare spatial models, ironically even in studies
which aimed to assess the presence of under- or over-smoothing (see for example, Rodrigues
and Assunção [25] and Law [26]). The problem is that these criteria are designed to quantify
goodness-of-fit (GoF), that is, the discrepancy between the observed data and the predicted

values from the model, while penalising for over-fitting (model complexity), but they fail to account for the spatial dependencies [27] and the effect that spatial smoothing has on model fit. Put another way, the problem of model selection can be viewed as an optimisation problem with several competing objective functions: in addition to GoF, model parsimony and predictive capability, spatial models necessitate an additional objective function–"goodness-of-smoothing" (GoS). Hence not only should a model which under- or over-smooths be given less preference, but a model with an appropriate amount of smoothing should be preferred over a model without any smoothing, even though it is likely to have a poorer GoF to the observed data.

In the context of Bayesian spatial modelling, spatial smoothing is typically implemented through a prior distribution using spatial weights to define the spatial dependencies; see Cramb et al. [28] for a critical review of popular Bayesian spatial models. One of the most common prior distributions for spatial random effects (SREs) in a Bayesian spatial model is the intrinsic conditional autoregressive (ICAR) prior [19, 29]. The BYM model [11] makes use of the ICAR prior, but also includes unstructured (independent) SREs so that the estimated risks are smoothed towards a local mean as well as a global mean [26]. The two random effects are henceforth referred to as the structured (SSRE) and unstructured spatial random effects (USRE). In response to the complexity of having two sets of SREs, Leroux et al. [30] proposed a model in which the SREs were a weighted mixture of the USRE and SSRE, the latter modelled by the ICAR prior. Although the BYM and Leroux models remain popular, especially in epidemiology [25], some concern about the potential for over-smoothing has been expressed (for example, see Smith et al. [19]; Law [26]; Kandhasamy and Ghosh [31], Lawson and Clark [32], Best et al. [33] and Cramb et al. [28])

This paper has three aims: 1) to demonstrate that reliance on common GoF criteria for spatial model selection is inadequate; 2) to propose several methods for quantifying the degree of smoothing; and 3) to compare these methods against GoF statistics on real data. These methods were developed within the context of disease-mapping using areal data in a Bayesian framework. However, some of these methods were inspired from methodology outside this field and will equally be applicable to problems in other contexts, such as geostatistics; other methods are more specific to the disease-mapping context, but could potentially be extended to a broader class of models and problems with little modification.

Without loss of generality, we impose three constraints on our study. The first is the range of models considered. We limit our analysis to the BYM and Leroux models for several reasons: they are well known and widely used; the ICAR model, which underpins both the BYM and Leroux models, has been criticised for being susceptible to over-smoothing; and as the ensuing analysis reveals, a wide range of models with varying degrees of smoothing can be achieved simply via changes to the hyperprior specification. For the purpose of quantifying and comparing different degrees of smoothing, this is adequate. Moreover, given the large influence of the hyperpriors on smoothing, the choice of model seems secondary. More broadly, other approaches such as models based on Gaussian process priors will suffer similar issues with respect to under- and over-smoothing.

The second constraint is investigating the effect of spatial smoothing parameters or spatial weights on the degree of smoothing. Typically, in models such as the BYM and Leroux, spatial weights are based on first-order adjacency. That is, each pair of spatial units (areas) are assigned a weight of 1 if they are considered (typically geographically) adjacent and zero otherwise. This simplifies the spatial covariance function substantially and improves computation without substantial loss of information. However, many other formulations have been explored (see for example Earnest et al. [27], Law [26], and Duncan et al. [34]). Not only has

this issue already received much attention, but the conclusions suggest that binary first-order adjacency weights are often a good choice anyway.

Third, the task of trying to determine the optimal amount of smoothing for a given model is not considered. Again, this has already been addressed in the literature (e.g. Evers et al. [35]), but more importantly, this task is impeded by the lack of guidance on how the degree of smoothing can be quantified.

The structure of this paper is as follows. The Methods section describes the Bayesian spatial models and introduces an important quantity derived from the model parameters which is subsequently used in the analysis. Also described in this section are five approaches to quantifying smoothing and three commonly used GoF measures, as well as the two spatial datasets. The Results section reports the parameter estimates, the GoF and GoS criteria are evaluated which are subsequently used to compare the models. These results and limitations of this study are examined in the Discussion.

## Methods

### Bayesian spatial models

For specificity, we consider two spatial models for area-level count data that are commonly used in epidemiological modelling. For each model, the data are assumed to follow a Poisson distribution

$$y_i \sim \text{Pois}(E_i e^{\mu_i})$$

where $y_i$ and $E_i$ are the observed and expected counts respectively, and $\mu_i$ is the log relative risk for the $i^{\text{th}}$ area. Assuming $k$ covariates and some weakly informative priors, the Leroux model [30] is specified as

$$\mu_i = \boldsymbol{\beta}^{\text{T}} \boldsymbol{x}_i + s_i$$

$$\beta_k \sim \mathcal{N}(0, \ \sigma^2)$$

$$s_i | \boldsymbol{s}_{\backslash i} \ \sim \ \mathcal{N}\left( \frac{\rho \sum_j w_{ij} s_j}{\rho \sum_j w_{ij} + 1 - \rho}, \ \frac{\sigma_s^2}{\rho \sum_j w_{ij} + 1 - \rho} \right)$$

$$\rho \sim \text{Unif}(0, 1)$$

$$\sigma_s^2 \sim \mathcal{IG}(\alpha, \ \eta)$$

and the BYM model [11] is specified as

$$\mu_i = \boldsymbol{\beta}^{\text{T}} \boldsymbol{x}_i + s_i + u_i$$

$$\beta_k \sim \mathcal{N}(0, \ \sigma^2)$$

$$s_i | \boldsymbol{s}_{\setminus i} \sim \mathcal{N}\left( \frac{\sum_j w_{ij} s_j}{\sum_j w_{ij}}, \frac{\sigma_s^2}{\sum_j w_{ij}} \right)$$

$$u_i \sim \mathcal{N}(0, \sigma_u^2)$$

$$\sigma_s^2 \sim \mathcal{IG}(\alpha, \eta)$$

$$\sigma_u^2 \sim \mathcal{N}(0, 10)^+$$

where $\boldsymbol{\beta}^{\mathrm{T}} = (\beta_0, \ldots, \beta_k)^{\mathrm{T}}$ are the k + 1 regression coefficients, $\mathcal{IG}$ denotes the inverse-gamma (IG) distribution, parameterised in terms of shape and rate, $\mathcal{N}(\cdot)^+$ denotes a Normal distribution left-truncated at zero, and all Normal distributions including the truncated distribution are parameterised in terms of mean and variance. The spatial weights $w_{ij}$ were fixed *a priori* as the binary, first-order adjacency weights, $\sigma^2$ was held fixed at 100, while different combinations of values of $\alpha$ and $\eta$ were used to fit different models with varying degrees of smoothing.

Given the sensitivity to the hyperprior for $\sigma_s^2$, left-truncated Normal (LTN) distributions, $\mathcal{N}(\pi, v)^+$, were also trialled. Other hyperpriors are possible (see Gelman [36] for example), but are not considered here for the sake of brevity. The specific values of $\alpha$, $\eta$, $v$, and $\pi$ are included in S1 Table. It should be stressed that these values are not necessarily sensible from a practical standpoint–they were chosen deliberately to induce a set of maps with varying degrees of smoothing to test the methods for quantifying smoothing described below. This yields a total of 4 models each with 12 model variants labelled A through L. While the relationship between the informativeness of a prior distribution and the impact it will have on smoothing is not straightforward, these model variants are approximately ordered in descending order of smoothing intensity.

**Extensions of the standardised incidence ratio.**   In the disease mapping context, the ratio $y_i/E_i$ is called the observed or 'raw' standardised incidence ratio (SIR). This is usually unstable due to low incidence and/or small populations at risk [10, 30, 37], and thus the goal is to provide a better estimate, given by the relative risk $\exp(\mu_i)$, or posterior SIR.

We introduce a new quantity, the covariate-adjusted SIR (CASIR), which is a key component of the methods below,

$$CASIR_i = \exp(\mu_i - \boldsymbol{\beta}^{\mathrm{T}} \boldsymbol{x}_i)$$

which is equivalent to exponentiating the SRE, $\exp(s_i)$. In the case of the BYM model, the unstructured spatial random effects (USRE) are also subtracted from $\mu_i$ before exponentiating. Similarly, we define the covariate-adjusted raw SIR (CARSIR) as

$$CARSIR_i = \frac{y_i}{E_i} \exp(-\boldsymbol{\beta}^{\mathrm{T}} \boldsymbol{x}_i)$$

where $y_i/E_i$ is the raw SIR. As will become apparent, the smoothed SIR surface, given by exp $(\mu_i)$, may not necessarily appear smooth, and paradoxically may appear less smooth when more smoothing is applied, and vice versa. This is because the smoothness exhibited by the SIR depends on the effect of the covariate(s), and their relative contribution to the SIR compared to the SRE. Conversely, the CASIR directly reflects the degree of smoothing.

We justify use of the CASIR over the SRE for two reasons. First, the CASIR is comparable to the SIR, the main parameter of interest in these epidemiological models, by converting the

SRE to a ratio scale parameter. Second, it allows a theoretical bound on the potential values of CASIR to be computed, which is a central feature of one of the approaches to quantifying smoothing described below. Taking logarithms of the raw SIR to compute a range for $s_i$ is not reliable since $y_i$ may be zero.

**Computation.** The Leroux model with the IG prior distribution was fit using the R package `CARBayes` [38], for computational efficiency while the other three models were fit using WinBUGS [39] via the R package `R2WinBUGS` [40, 41]. Although `CARBayes` can fit the BYM model with an IG prior, only the sum of the estimated SREs are provided whereas separate estimates of the SSRE and USRE are highly valuable for this analysis. These software use Markov chain Monte Carlo (MCMC) techniques to estimate the posterior distribution. Although other software is available which should produce very similar results, these software were chosen for their reliability and convenience in fitting these particular spatial models.

## Approaches to quantifying smoothing

There are potentially several ways to quantify the degree of smoothing attained by a given model. To address the second aim of this paper, five ideas are explored. The origins of these ideas and their technical details are described below.

**Ratio of variograms.** The classical variogram for area $i$ at lag $h$ is given by

$$\gamma_i(h) = \frac{1}{2N_i(h)} \sum_{j \sim i} (z_i - z_j)^2$$

where $N_i(h)$ is the number of areas which are no more distant than the lag $h$ from area $i$, and $j \sim i$ denotes all areas $i$ and $j$ which satisfy $d_{ij} < h$ where $d_{ij}$ is the distance between areas $i$ and $j$, and $z_i$ is a measured variable for area $i$ [3, 6]. Instead of using the Great Circle distance between the centroids of each area, we define $d_{ij}$ as the minimum number of boundaries that must be crossed to move from area $i$ to area $j$, as proposed by Knorr-Held and Raßer [42]. This appears to be more appropriate for areal data as it tends to provide smoother and more robust estimates of the variogram, especially for small lag values. Additionally, under this construction, adjacency of areas defines the autocorrelation in the variogram as well as the weights matrix in the modelling.

The variogram, averaged over the areas,

$$\gamma(h) = \frac{1}{N} \sum_{i=1}^{N} \gamma_i(h),$$

provides a succinct visual representation of the spatial continuity of the variable $\mathbf{z} = (z_1, \ldots, z_N)$. Plotting the variogram of CASIR against the variogram of the CARSIR may be helpful in assessing the degree of smoothing: a variogram that is too flat indicates over-smoothing, while a variogram that is similar to that for the raw SIR indicates under-smoothing. As a quantitative metric for assessing GoS, we propose the ratio of the variograms for CASIR to CARSIR, averaged over the areas and lag parameter. This can be compared against a user-specified target to determine whether the smoothing is appropriate.

**Kurtosis preservation.** Drawing on inspiration from developments in time series analysis, we propose a method based on the work of Rong and Bailis [43]. The authors address the issue of over-smoothing in time series analysis by using a simple moving average smoothing function such that the moving average window size minimises the "roughness" (defined as the standard deviation of the first-order difference series) with the constraint that the kurtosis of the smoothed time series must be greater than or equal to the kurtosis of the original,

unsmoothed time series. That is, they aim to smooth a time series as much as possible while preserving kurtosis. The result is that the smoothed time series retains rare large-scale deviations while smoothing out more frequent modestly sized deviations.

This methodology presented in Rong and Bailis [43] not only provides a technique for smoothing, but also a statistic for quantifying smoothness. It is the latter development that is of interest here, since the spatial smoothing is performed as part of the Bayesian modelling. However, spatial dependencies differ from longitudinal dependencies in terms of how individual units (areas or time points) are assumed to interact. As an analogy to first-order differences in time, we consider a first-order neighbourhood approach in space, that is, differences between a measure at a given area and the mean of its first-order neighbours. The roughness is the standard deviation of these differences over all areas.

For a generic spatial variable $z_i$ associated with the $i^{\text{th}}$ area, the excess kurtosis is defined as

$$\text{Kurt}(z_i) = \frac{\mathbb{E}[(z_i - \bar{z}(\boldsymbol{w}_i))^4]}{\mathbb{E}[(z_i - \bar{z}(\boldsymbol{w}_i))^2]^2} - 3$$

where $\bar{z}(\boldsymbol{w}_i)$ is the weighted mean of $\{z_i; i = 1, \ldots N\}$, and $\boldsymbol{w}_i$ is the vector of spatial weights pertaining to the $i^{\text{th}}$ area. The overall measure of kurtosis is given by averaging over all areas, $i = 1, \ldots, N$. A larger kurtosis implies that the variation is dominated by infrequent and extreme deviations [43].

Note that whether $\boldsymbol{z}$ is defined as the CASIR or SIR, the kurtosis is very similar when compared with their raw counterpart (i.e. CARSIR and raw SIR). However, the roughness can vary substantially, making inference difficult. In our analyses, the SIR was found to be a more reliable measure, which is what is presented here. For consistency, SIR was also used to compute the kurtosis, i.e. $z_i = SIR_i$.

**Kappa.** Cohen's kappa statistic [44] has been used previously in the spatial context to compare spatial agreement of patterns and to quantify the magnitude of spatial smoothing (e.g. Sterlacchini et al. [45] and Earnest et al. [27]). The statistic is defined as

$$\kappa = \frac{\Pr(A_o) - \Pr(A_e)}{1 - \Pr(A_e)}$$

where $\Pr(A_o)$ and $\Pr(A_e)$ are the observed and expected proportion of agreement between a spatial variable respectively,

$$\Pr(A_o) = \frac{1}{N} \sum_{i=1} c_{ii}$$

$$\Pr(A_e) = \frac{1}{N} \sum \frac{\sum_{j=1} c_{ij} \times \sum_{i=1} c_{ij}}{N}$$

and $\{c_{ij}\}$ are the elements of a confusion matrix formed from the cross-tabulation of the categories of nominal variables [44]. To cross-tabulate values of continuous variables like the observed and smoothed SIRs, they must first be categorised by specifying "epidemiologically meaningful" thresholds [27, 46]. Following the suggestions of Earnest et al. [27] and Sterlacchini et al. [45], kappa was computed on the quantiles of CASIR and CARSIR using 3 categories (2 cut-offs: 0.25 and 0.75) as well as 5 categories (4 cut-offs: 0.1, 0.3, 0.7, 0.9).

In addition to being designed for categorical data, Cohen's kappa has several criticisms. Interpretation of kappa is not straightforward since its magnitude can be influenced by multiple factors, and it may not be clear which factor(s) is responsible [45, 46]. While there is no

consensus to interpreting kappa, some guidelines have been suggested in the literature (e.g. Landis and Koch [47]). Broadly, kappa values less than or close to zero indicate a lack of agreement, while kappa values close to 1 indicate substantial agreement [45–47]. However, the difficulty of interpreting kappa is exacerbated in the spatial context. The statistic does not take into account the spatial structure of the two variates being compared, and being symmetric, there is no clear "baseline" for assessing agreement. Consequently, there is no unambiguous connection between kappa and the degree of smoothness exhibited by the spatial variables. This problem is illustrated in Fig 1.

In Fig 1A and 1B, there is perfect agreement between variables A and B. However, the surfaces in b) are not smooth, so a kappa value close to 1 does not necessarily indicate a high degree of smoothness. In Fig 1C, there is perfect disagreement, yet both surfaces are smooth, so the low kappa value should not be interpreted as a low degree of smoothness. In Fig 1D, kappa is approximately 0.04. Regardless of how this is interpreted, it is not clear how it would apply to surfaces A and B simultaneously.

If one of the two variables being compared is designated as the baseline, then this may help in the interpretation. For example, consider the two variables raw and smoothed SIR. The null hypothesis is that the raw SIR is not smooth. As smoothing increases, the disagreement between these variables will increase, thereby reducing the kappa value. Thus it has been suggested that smaller kappa values indicate greater smoothing [27].

In the absence of more definitive guidelines, the following metric to assess the GoS was devised using the results from the other methods as calibration: $\hat{\kappa} < 0.05$ indicates over-

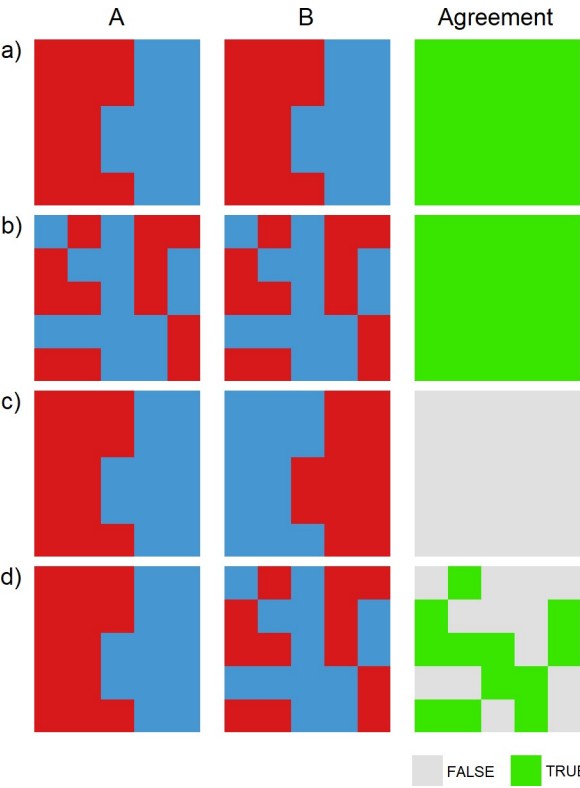

**Fig 1. Examples of Cohen's kappa for 2 spatial variables categorised into 2 groups, illustrating the difficulty in interpretation of kappa with respect to degree of spatial smoothing.** The kappa values are a) $\hat{\kappa} = 1$, b) $\hat{\kappa} = 1$, c) $\hat{\kappa} = -1$, and d) $\hat{\kappa} \cong 0.04$.

smoothing, $0.05 < \hat{\kappa} < 0.95$ indicates a reasonable degree of smoothing, and $\hat{\kappa} > 0.95$ indicates under-smoothing.

Note that Earnest et al. [27] compute kappa for the raw and smoothed SIR. However, the only covariates included in their models are temporal, not spatial, making these variables more comparable to the CARSIR and CASIR respectively. As explained above when introducing CASIR, it is necessary to remove the effect of spatial covariates when assessing spatial smoothing. Consequently, in this paper, kappa is computed for the estimates of CASIR and CARSIR, treating the latter as the baseline for agreement.

**Fraction of spatial variation.** Earnest et al. [27] and Law [26] also consider comparing models based on the fraction of spatial variation explained by the model. This is defined as the ratio of the empirical variance captured by the SSRE to the total spatial variation,

$$\psi = \frac{\text{Var}(\boldsymbol{s})}{\text{Var}(\boldsymbol{s}) + \text{Var}(\boldsymbol{u})}$$

where $\boldsymbol{s}$ and $\boldsymbol{u}$ are the SSRE and USRE in the BYM model respectively–the only model considered by Earnest et al. [27]. As illustrated in Duncan et al. [34], this ratio, albeit using standard deviations rather than variances, is helpful in solving the identifiability issue between $\boldsymbol{s}$ and $\boldsymbol{u}$, by modifying these random effects according to $\psi$, which has been applied to the results from all the BYM model variants in this paper. It is not meaningful to compute this ratio again after modification, nor is this ratio applicable to other models which have only one set of SREs, like the Leroux model.

To generalise this concept to all spatial models with a SRE, $\boldsymbol{s}$, we propose redefining the total spatial variation to be $\text{Var}(s) + \text{Var}(\varepsilon)$ where $\boldsymbol{\varepsilon} = (\varepsilon_1, \dots, \varepsilon_N)^{\text{T}}$ are the model residuals, which for the BYM model includes the unstructured spatial random effect. That is, the residuals for the BYM model are defined as

$$\varepsilon_i = E_i e^{\mu_i - u_i} - y_i$$

since the USREs do not contribute to an understanding of the spatial variation but rather represent spatial noise. To compute the ratio, the posterior median for a posterior sample of size $M$ is computed before computing the variance over the areas, i.e.

$$\text{Var}(\boldsymbol{s}) = \frac{1}{N-1} \sum_{i=1}^{N} (s_i^* - \bar{s}^*)^2$$

$$\text{Var}(\boldsymbol{\varepsilon}) = \frac{1}{N-1} \sum_{i=1}^{N} (\varepsilon_i^* - \bar{\varepsilon}^*)^2$$

where

$$s_i^* = \underset{m=1,\dots,M}{\text{median}} \{s_i^{(m)}\}$$

$$\bar{s}^* = \frac{1}{N} \sum_{i=1}^{N} s_i^*$$

and similarly for $\varepsilon_i^*$ and $\bar{\varepsilon}^*$. Whether a small or large fraction of spatial variation is preferred depends on the reason for modelling the SIR [27]. Moreover, it is not obvious what values would be considered small or large in general or in a particular application. Given the lack of guidelines for interpreting this statistic for the purpose of assessing the degree of spatial

smoothing, this criterion was not given further consideration when comparing the models. However, the results are reported below for completeness.

**Relative position of CASIR.**   The fifth approach to quantifying smoothing begins with the observation that if no smoothing (i.e. no shrinkage) occurs, then the smoothed SIR and *CASIR* become more similar to their raw counterparts. As the degree of smoothing increases, each estimate of the SIR is smoothed towards the mean of its neighbours, subject to the model constraints and *a priori* knowledge imposed by the prior distributions. When the maximum amount of smoothing is applied to area *i*,

$$CASIR_i \rightarrow \text{E}(CASIR_{j \sim i}|\boldsymbol{y})$$

which approaches 1 as the SRE tends to zero. This does not imply that all areas will be smoothed towards the global mean, since areas may experience different degrees of smoothing. In fact, some areas will undoubtedly be smoothed *away* from the global mean. Notwithstanding some small deviations due to the use of posterior point estimates and properties of the posterior sample such as convergence and effective sample size, the $CASIR_i$ estimate will lie somewhere between $CARSIR_i$ and the posterior mean of its neighbours, $\text{E}(CASIR_{j \sim i}|\boldsymbol{y})$. If the relative position of CASIR at these two extremes is denoted 0 and 1 respectively, then this quantifies the degree of smoothing exhibited by a given area in relative terms. To quantify the overall degree of smoothing for a given model, the distribution of these relative positions is compared against a specified cut-off (see Table 1 for some examples).

**Assessing GoS criteria.**   Several criteria were used to classify the models based on example cut-offs, listed in Table 1. These cut-offs can be adjusted in the same way that different cut-offs for DIC and WAIC can be specified to broaden or narrow the set of models considered "good". A "PASS" indicates that the model variant is neither under- nor over-smoothing under the given criterion. Note that unlike the other GoS approaches, the kurtosis preservation method only has 2 cut-offs as it is not obvious how this criteria can be adjusted to penalise

**Table 1. Cut-offs used to construct the GoS criteria.**

| Statistic | Cut-off type | Criteria |
|---|---|---|
| Variogram ratio | (u) | The ratio, averaged over the lag, is between 0.2 and 0.8. |
| | (c) | The ratio, averaged over the lag, is between 0.25 and 0.75. |
| | (pu) | The ratio, averaged over the lag, is between 0.1 and 0.4. |
| Kurtosis Preservation | (u) | The kurtosis of CASIR ≥ kurtosis of CARSIR and the roughness of CASIR is less than the minimum roughness + 30% |
| | (c) | The kurtosis of CASIR ≥ kurtosis of CARSIR and the roughness of CASIR is less than the minimum roughness + 10% |
| Kappa | (u) | Kappa lies between 0.05 and 0.95. |
| | (c) | Kappa lies between 0.1 and 0.9. |
| | (pu) | Kappa lies between 0.05 and 0.7. |
| Relative position of CASIR | (u) | At least 75% of the *N* CASIR point estimates lie within the range 0.01 to 0.99 (inclusive). |
| | (c) | At least 85% of the *N* CASIR point estimates lie within the range 0.02 to 0.98 (inclusive). |
| | (pu) | At least 75% of the *N* CASIR point estimates lie within the range 0.2 to 0.98 (inclusive). |

(u) = unbiased; (c) = conservative (less likely to choose under- or over-smoothed mode ls); (pu) = penalise under-smoothing more heavily than over-smoothing.

under-smoothing in favour of models with more smoothing. For the reasons outlined above, the fraction of spatial variation is excluded.

## Goodness-of-fit and predictive performance

To address the first and third aims of this paper, we consider the following criteria commonly used to measure GoF and check predictive performance. Many studies involving model selection amongst competing spatial models use DIC which evaluates the model GoF while penalising for model complexity (e.g. Law [26] amd Earnest et al. [27]). The DIC was proposed by Spiegelhalter et al. [23] as a generalisation of Akaike's information criterion (AIC) [48] using information theoretic justification. The DIC can be defined as

$$DIC = 2p_D - 2\log p(\boldsymbol{y}|\bar{\boldsymbol{\theta}})$$

where $p_D$ is the effective dimension of the model and $p(\boldsymbol{y}|\bar{\boldsymbol{\theta}})$ is the likelihood evaluated at the posterior mean of the unknown parameters, $\boldsymbol{\theta}$. The WAIC [24, 49] is a similar criterion, defined as

$$WAIC = 2p_W - 2\log \prod_{i=1}^{N} \mathbb{E}_{\theta}[p(y_i|\theta_i)|y_i].$$

The advantages of WAIC over DIC include that it uses the entire posterior distribution, is invariant to parameterisation, and closely approximates Bayesian cross-validation [49, 50]. Both GOF criteria are considered here for comprehensiveness.

Gelman et al. [49] propose two variants of $p_W$. Here we use the second variant,

$$p_W = \frac{1}{2} \sum_{i=1}^{N} \text{var}[\log p(y_i|\theta_i)|y_i]$$

which, after simplification, leads to the specific WAIC criterion

$$WAIC = 2 \sum_{i=1}^{N} \left\{ \underset{m=1,\ldots,M}{\text{var}} [\log p(y_i|\theta_i^{(m)})] - \log \frac{1}{M} \sum_{m=1}^{M} p(y_i|\theta_i^{(m)}) \right\}$$

where $\theta_i^{(m)}$ is the estimate of the unknown parameter(s) for the $i^{\text{th}}$ area and $m^{\text{th}}$ MCMC iteration. Predictions, or theoretical future observations, denoted $\tilde{y}$, can be drawn from the posterior predictive distribution

$$p(\tilde{\boldsymbol{y}}|\boldsymbol{y}) = \int p(\tilde{\boldsymbol{y}}|\boldsymbol{\theta})p(\boldsymbol{\theta}|\boldsymbol{y})d\boldsymbol{\theta}$$

which can be used to assess predictive performance. The idea is that if the model is adequate in describing the data generating process, then the predicted data $\tilde{y}$ will be close to the observed data $\boldsymbol{y}$. Thus these posterior predictive checks (PPCs) can be viewed as a variation on GoF diagnostics [51].

One specific PPC is the conditional predictive ordinate (CPO) [52] which seeks to re-observe a datum $y_i$ given all other observed data, denoted $\boldsymbol{y}_{/i}$,

$$CPO_i = p(y_i|\boldsymbol{y}_{\backslash i})$$
$$= \int p(y_i|\boldsymbol{\theta})p(\boldsymbol{\theta}|\boldsymbol{y}_{\backslash i})d\boldsymbol{\theta}.$$

This metric is equivalent to the posterior predictive ordinate (PPO), $p(y_i|y)$, in the sense that the set of leave-one-out marginal distributions $\{p(y_i|y_{\backslash i}); i = 1, \ldots, N\}$ contain the same

information about the predictive performance as the marginal distribution p($y$) [51, 53]. However, the CPO avoids double use of the data since it is a leave-one-out cross-validation predictive density. Additionally, unlike the PPO, the literature contains several useful guidelines for interpreting the CPO [51]. For detecting outlying observations, Congdon [54] suggests scaling the CPO values by dividing them by the maximum CPO value. Scaled CPOs less than 0.01 suggest areas for which the model does not fit well. To compare models, several overall measures of fit have been proposed (e.g. Ntzoufras [51] and Congdon [54]). However, the most numerically stable option seems to be the sum of the log CPO values, as suggested by Held et al. [55], which we adopt here. The best model is taken to be the model which minimises

$$-\sum_{i=1}^{N} \log(CPO_i).$$

In addition to these GoF criteria, we use Moran's I statistic [56] to measure the degree of autocorrelation remaining in the model residuals, checking the model assumption that the residuals are independent and identically distributed.

To compare models with respect to predictive performance, the minimum DIC and WAIC were determined for each model, indicating the best model fit, and model variants with a DIC or WAIC within 2 or 7 units were identified as having reasonable model fit, as per the common rule of thumb [23]. Smaller sums of log CPOs indicated better predictive performance, and the model with the minimum was flagged. Moran's I was compared across model variants using p-values from the test assuming normality of the statistic under the null hypothesis of no autocorrelation.

## Data

Two spatial datasets are analysed. The first is the North Carolina sudden infant death syndrome (SIDS) dataset first presented by Atkinson [57], and subsequently augmented and analysed by Cressie and Read [37] and Cressie and Chan [58] amongst others. The observed data represent counts of SIDS aggregated from 1979 to 1983 for each of the 100 counties in North Carolina. The non-white birth rate over the same period is included here as a covariate. The second dataset is the Scottish lip cancer dataset compiled by Kemp et al. [59] and first analysed in Clayton and Kaldor [9]. This data has been previously analysed by Spiegelhalter et al. [23], Leroux et al. [30], and Duncan et al. [34] amongst others. The observed data represent counts of lip cancer across 56 counties of Scotland, and a spatial covariate representing the percentage of the workforce acting as a proxy for sun exposure is included. A graphical summary of the data is shown in Fig 2. To improve visual interpretation, the northeast island counties of Scotland, Shetland and Orkney, are excluded from all maps. This modification is limited to the maps–data from these counties are still used and estimates for these counties are still generated by the models.

These datasets were chosen for the following reasons: they each contain one useful spatial covariate, which is essential in demonstrating the importance of CASIR; each study region contains a sufficient number of areas to enable adequate evaluation of spatial effects; they have been extensively analysed previously, corroborating the plausibility of the model specifications and parameter estimates presented here; and they are publicly available data, facilitating reproducibility. Additionally, these data represent real cases. This has the advantage over simulated data which may not resemble realistic data, thus casting doubt on the authenticity of the model results and accuracy of the approaches to quantifying smoothing.

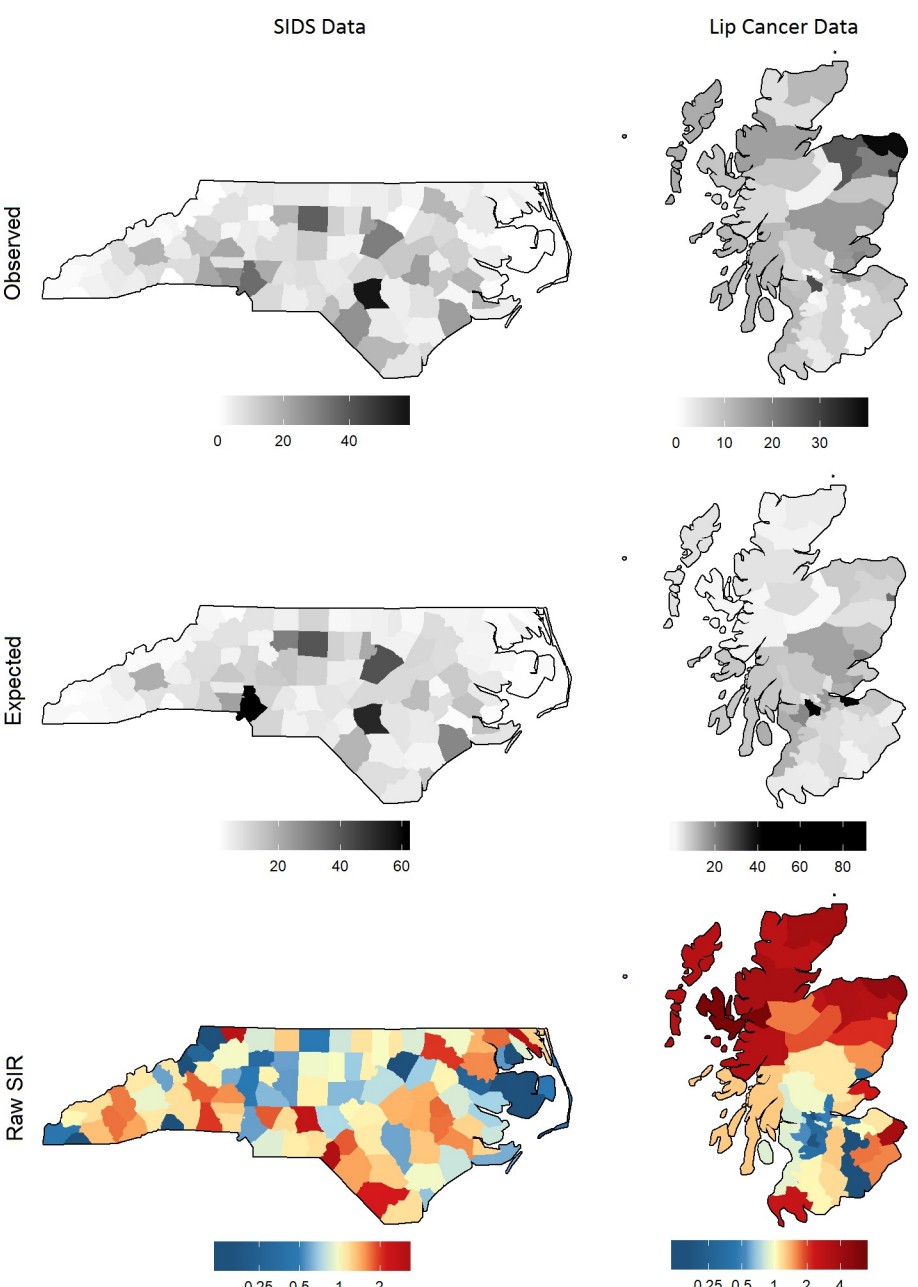

**Fig 2. Summary of the two datasets.** The observed and expected counts are shown in greyscale; the gradient is capped at the maximum observed value (57 and 39 respectively); larger expected values are shown in black. The colour gradient for the raw SIR reflects a ratio scale; darker shades of red indicate a higher raw SIR, while darker shades of blue indicate a lower raw SIR.

## Results

For the sake of brevity, the ensuing figures relate mostly to the lip cancer data, with the remaining results presented in S1 Appendix. Key parameter estimates for the BYM model with IG hyperpriors fit to the lip cancer dataset are summarised in Fig 3. The values represent the posterior means. The first four columns correspond to the linear scale parameters: the SSRE ($s_i$), USRE ($u_i$), covariate effect ($\beta x_i$), and the logarithm of the smoothed SIR ($\mu_i$). The last two

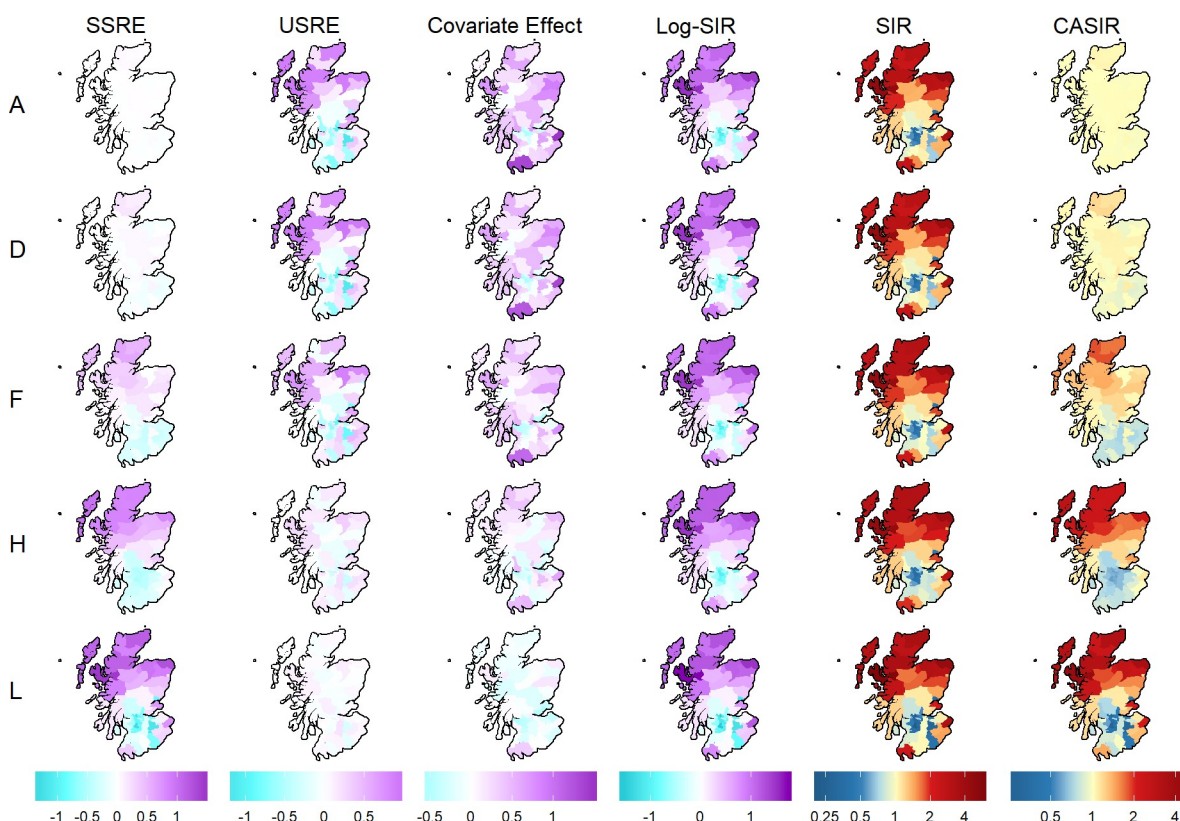

**Fig 3. Maps showing the posterior mean estimates of the key model parameters for 5 select model variants of the BYM model with an IG hyperprior (lip cancer dataset).**

columns correspond to the ratio scale parameters, namely the SIR ($e^{\mu_i}$) and CASIR ($e^{s_i}$). The colour gradient is consistent within each of these two classes of variables (i.e. same hues indicate the same values), but the legend reflects the range of values for the specific variable. Note that the degree of smoothing generally decreases as the model variant increases from A to L.

Maps of the key parameter estimates for all the alternative models and variants, for both the lip cancer and SIDS datasets are provided as supplementary material (see S1 Fig through S8 Fig).

The spatial pattern of the SIR appears similar across model variants, while the CASIR varies considerably. The contrast between the SIR and CASIR is greater when more smoothing is applied, highlighting the value of CASIR when trying to investigate the occurrence over-smoothing. This is particularly true for this model applied to this dataset, as the maps of the SIR look similar to the map of the raw SIR in Fig 2. A visual inspection of the SIR maps in Fig 3 (and maps of the remaining 7 model variants in S3 Fig) might lead one to conclude that all model variants have under-smoothed, when in fact the majority of the model variants are likely to be over-smoothed, as the subsequent analysis reveals. Moreover, aside from the SIR, these model variants vary considerably in the estimated SSRE, USRE, and covariate effect, each providing different statistical inference.

The smoothing paradox effect on the SIR surface is not readily observed in Fig 3, but is quite noticeable in the results for the Leroux model variants on the lip cancer data (see S1 and S2 Figs), and the BYM model variants on the SIDS data (see S7 and S8 Figs). The extent of this effect depends largely on the contribution of the covariate effect to the log-risk surface and how spatially autocorrelated the covariate is.

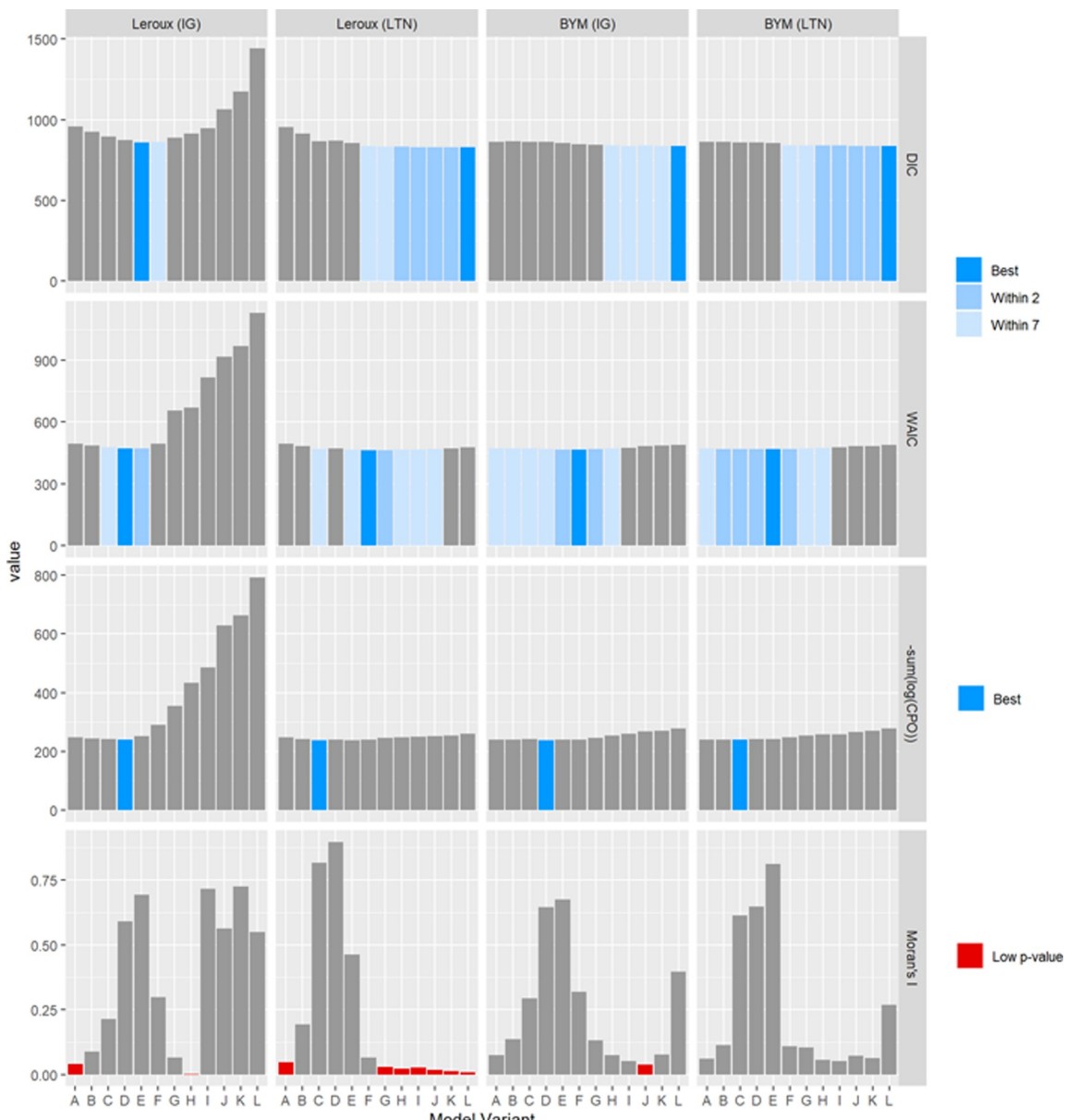

**Fig 4. Values of the GoF criteria and Moran's I p-values for each model variant fit to the SIDS dataset.**

### Goodness-of-fit criteria

The results for the GoF criteria and Moran's I p-values are summarised in Figs 4 and 5.

The interpretation of Figs 4 and 5 is the same. For the DIC and WAIC, the model that minimises the respective criterion is highlighted blue. This is the best model under this criterion. Models with a DIC or WAIC value within 2 or 7 units are highlighted in lighter shades of blue, indicating a reasonable model fit. For the CPO, the model which minimises the criterion is highlighted. For Moran's I, models with small p-values are highlighted red.

There are two important observations to be made here. First, the GoF criteria DIC, WAIC, and CPO are rarely in agreement, and sometimes identify very different models. For example, in Fig 4, for the Leroux LTN model, the model variants considered "best" under each of the

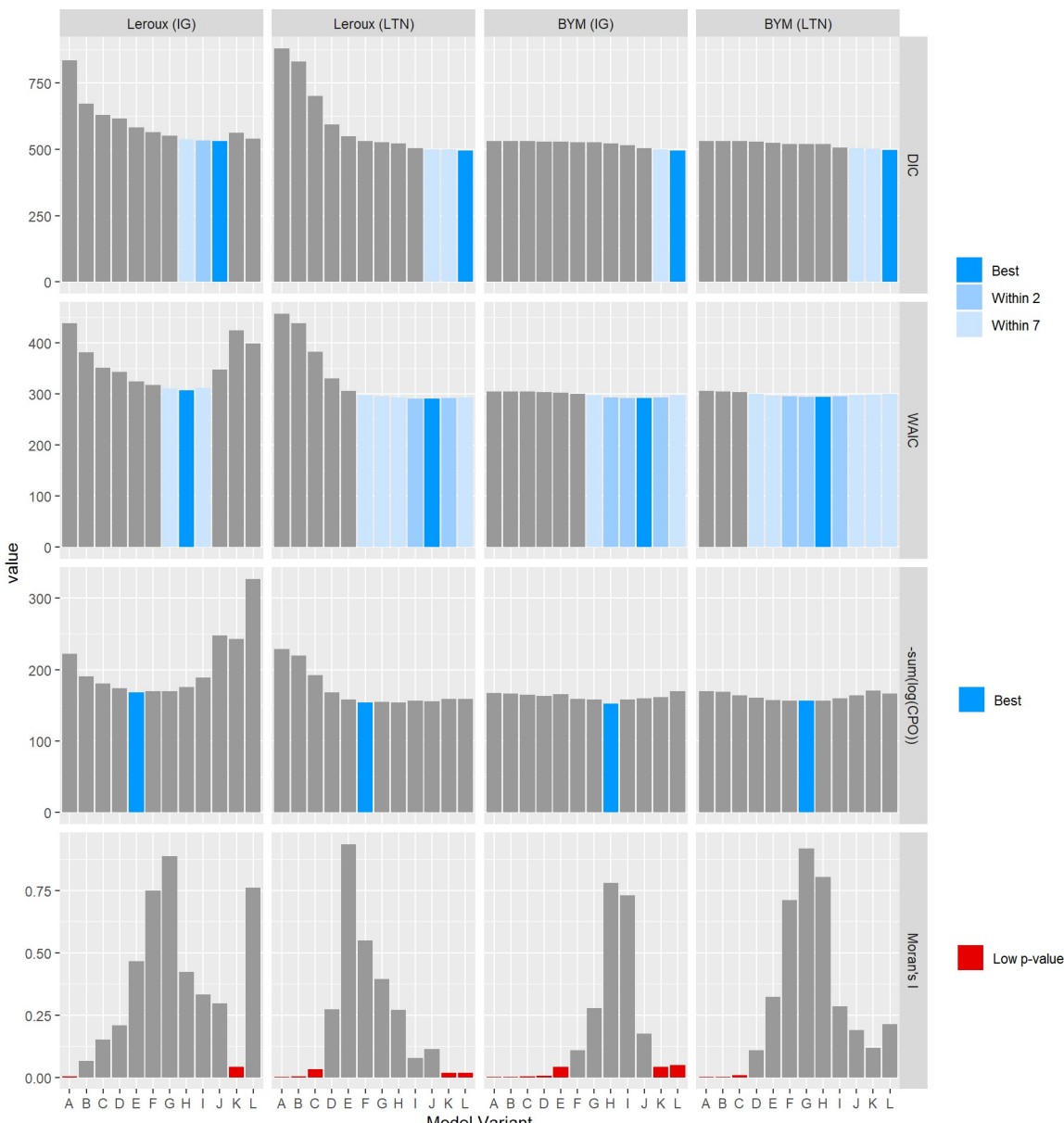

**Fig 5. Values of the GoF criteria and Moran's I p-values for each model variant fit to the lip cancer dataset.**

three GOF criteria are L, F, and C. Second, sometimes the best model under DIC coincided with a low Moran's I p-value. However, Moran's I p-values should be interpreted cautiously–a high degree of autocorrelation amongst near-zero residuals should not warrant the same concern as highly autocorrelated residuals that are large in magnitude. This is especially true for those models closer to variant L which have less smoothing and therefore generally have smaller residuals (see S9 and S10 Figs).

## Goodness-of-smoothing criteria

**Ratio of variograms.** The variograms for the lip cancer data are shown in Fig 6. Similar results hold for the SIDS dataset (see S11 Fig). In general, as the smoothing decreases, the

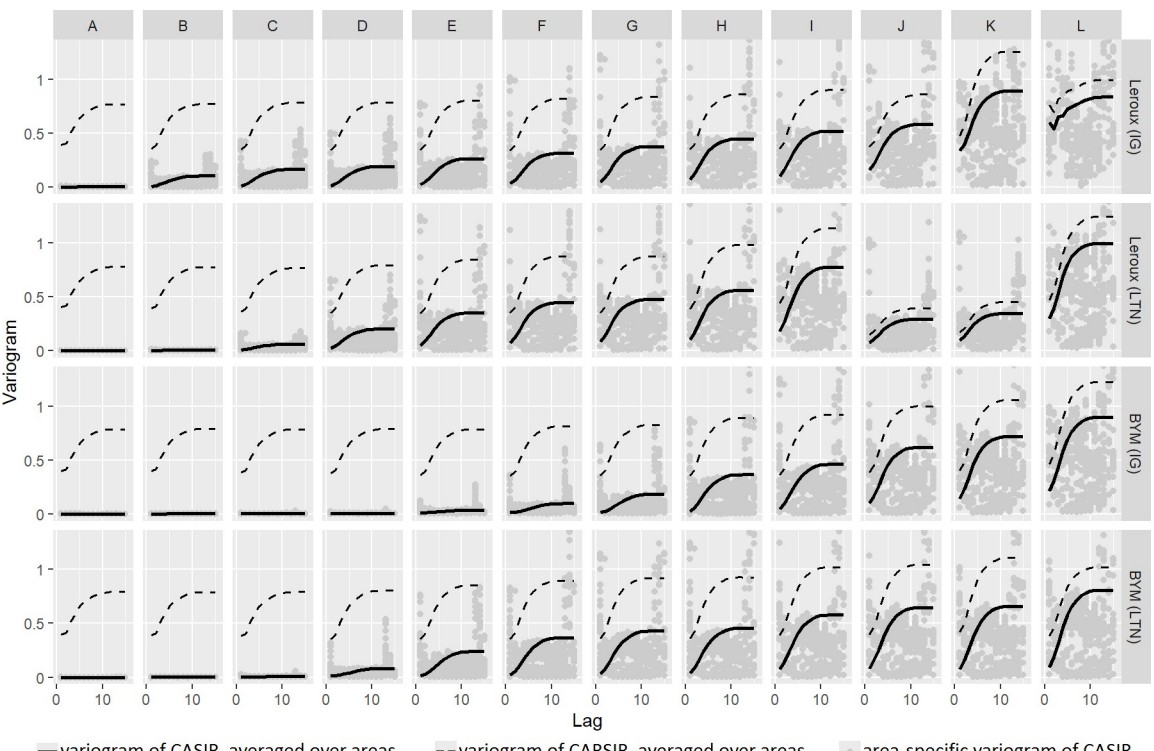

**Fig 6. Variograms for each model variant fit to the lip cancer dataset.** The solid and dashed lines denote the variograms of CASIR and CARSIR respectively, each averaged over the areas. The grey dots denote the area-specific variogram of CASIR. Note that the y-axis has been capped at 1.3 for clarity.

relative distance between the CASIR and CARSIR variograms decreases. That is, the ratio of the CASIR variogram to the CARSIR variogram, averaged over the lag, increases.

**Kurtosis preservation.** The kurtosis and roughness for the models fit to the lip cancer dataset are shown in Fig 7. Recall that the aim is to preserve the spatial kurtosis of the SIR with respect to the raw SIR while minimising the roughness of the SIR. The kurtosis was generally preserved for the Leroux models, and less frequently for the BYM models. The model variants in the middle (e.g. S4 Fig through S9 Fig) tend to have less roughness, steering model choice away from more extreme models which are likely to be over- or under-smoothing.

The results for the SIDS dataset (see S12 Fig) were less clear, with the SIR kurtosis values being less than the raw SIR kurtosis except for 5 model variants. This suggests that it is not only the type of model (i.e. Leroux vs BYM) that influences how well the kurtosis is preserved, but that it may also depend on other factors including characteristics of the data. Also contrary to the lip cancer data results, the roughness for the SIDS models generally increased with the model variants from A through L.

**Kappa.** The values of the kappa statistic for the lip cancer data, representing the spatial agreement between CASIR and the baseline CARSIR are shown in Fig 8. The values of kappa generally increase with model variant, as expected. There is not much difference between the kappa values whether 3 or 5 categories are used, but using fewer categories generally improves the robustness of this estimate since there is more information contributing to each cell of the confusion matrix. The results are generally similar for the SIDS data (see S13 Fig), although the kappa values start to decrease for some of the BYM model variants with less smoothing.

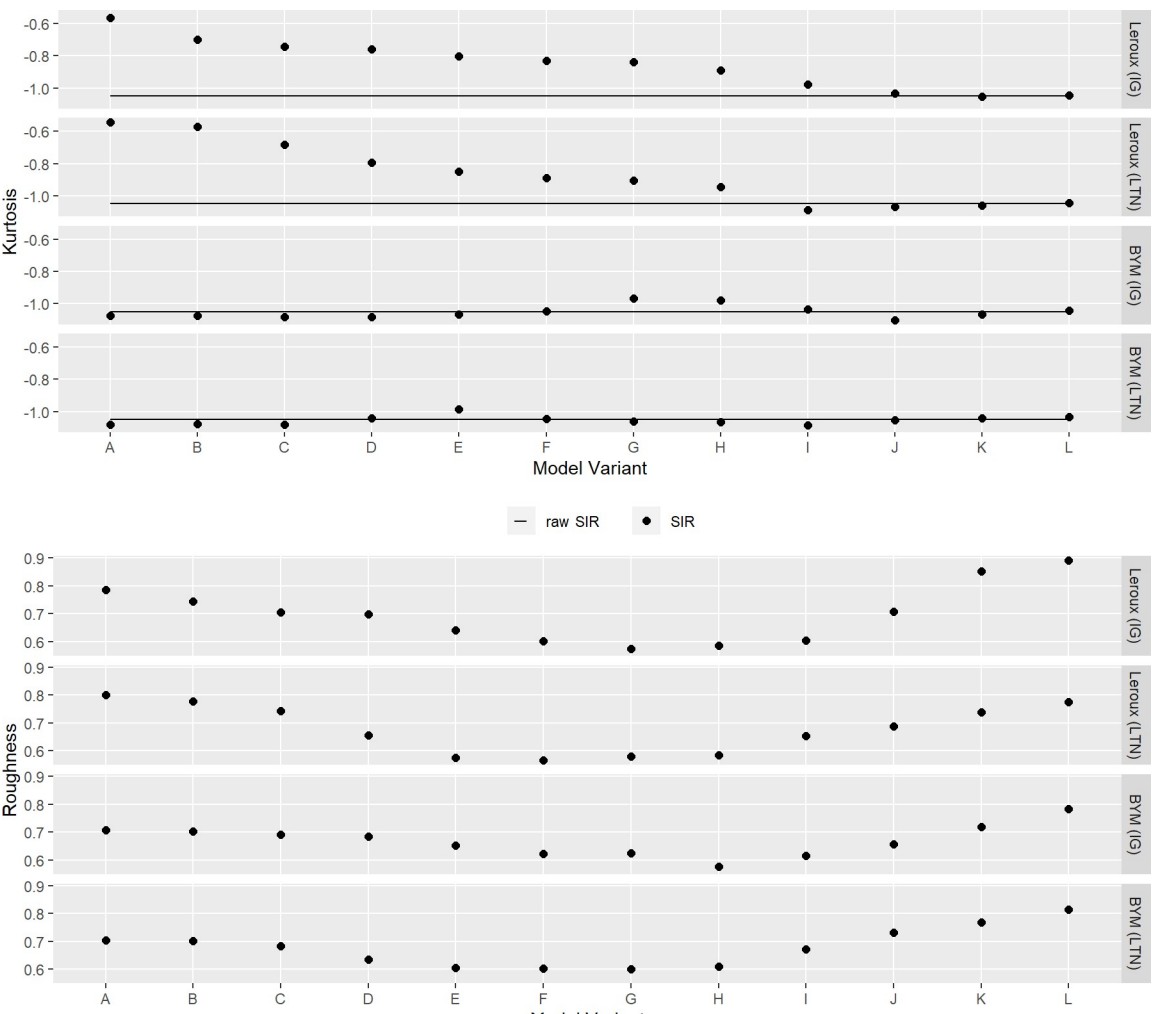

**Fig 7. Kurtosis and roughness for each model variant fit to the lip cancer dataset.** The horizontal lines denote the kurtosis of the raw SIR. The black dots denote the estimates of kurtosis and roughness.

**Fraction of spatial variation.** The results of the fraction of spatial variation for the lip cancer dataset are shown in Fig 9. The results for the SIDS dataset exhibit a similar trend and magnitude of values (see S14 Fig). For both datasets, the fraction of spatial variation ranges between 0% and 10% approximately, and generally increases with model variant, similar to the kappa statistic.

**Relative position of CASIR.** The posterior mean CASIR values and their relative position for select variants of the BYM IG model fit to the lip cancer dataset are shown in Fig 10. These results correspond to the maps shown in Fig 3. The mean CASIR estimates are denoted by the filled circles, which are situated within the range of potential values. When the degree of smoothing is large, these estimates tend to lie towards the end of the range representing the mean of the neighbouring values. As smoothing decreases, these estimates tend to move towards the opposite end of the range, representing the CARSIR estimates. Note that in general, as smoothing increases, the CASIR estimates are smoothed towards the global mean of 1. However, the direction a given estimate of CASIR moves is not necessarily towards 1; sometimes the CASIR estimate will be smoothed *away* from 1, depending on the neighbouring values.

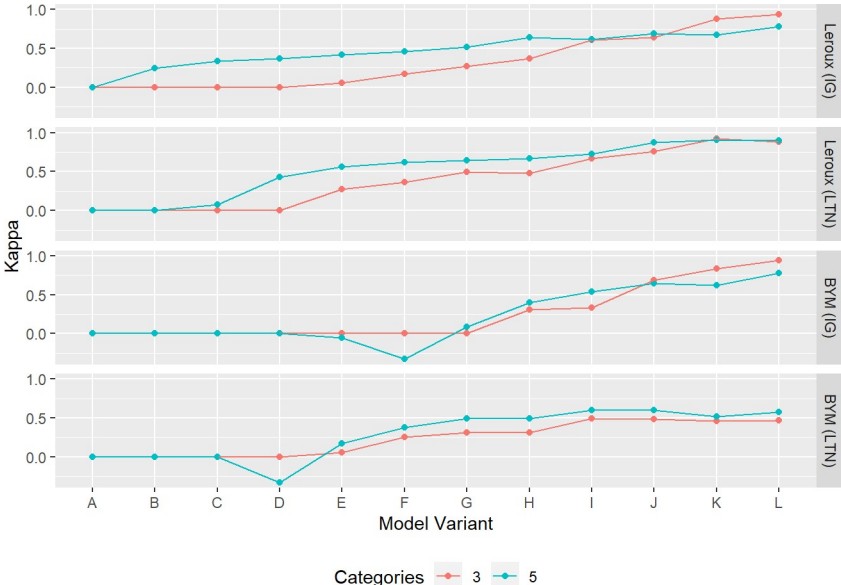

**Fig 8. Kappa statistic between CASIR and CARSIR for each model variant fit to the lip cancer dataset, using 3 and 5 discrete categories.** Values close to 0 suggest over-smoothing while values close to 1 suggest under-smoothing.

The CASIR values may lay outside the range of potential values due to the flexibility afforded by the prior distribution–the less informative the hyperprior for $\sigma_s^2$, the greater the propensity. This effect is minimised by taking the posterior mean of the CASIR values, but conversely, the effect is exaggerated when the range of potential values is very small, thus over-estimating the effect of under- or over-smoothing for these areas. To address this, the relative position of CASIR was not computed for areas when the logarithm of the range of potential values was less than 0.03.

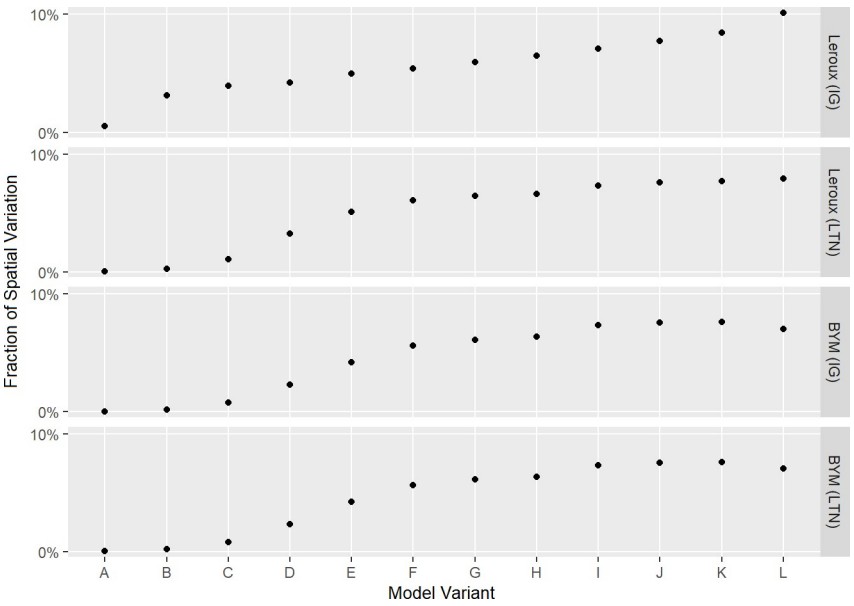

**Fig 9. Fraction of spatial variation for each model variant fit to the lip cancer dataset.**

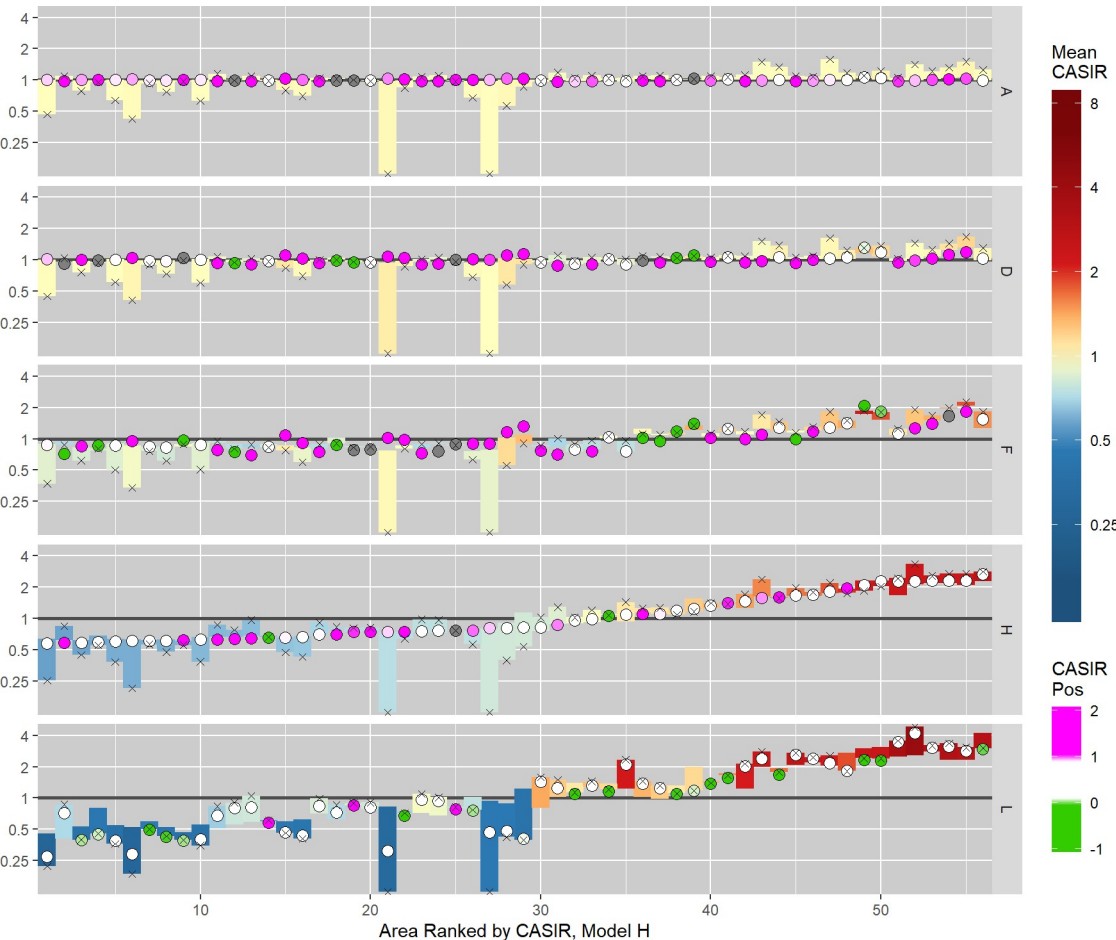

**Fig 10. Area-specific posterior mean estimates of CASIR and their relative position for select model variants of the BYM IG model (lip cancer data).** The coloured bars represent the theoretical range of CASIR values, coloured according the mean CASIR estimates, with the cross symbol marking the endpoint corresponding to the CARSIR estimate. The dots represent the mean CASIR estimate, coloured according to the relative position, with the pink and green colours indicating cases that are likely over- and under-smoothing respectively (grey indicates areas that were excluded due to the theoretical range of CASIR values being too narrow).

The distribution of the relative positions for each model variant fit to the lip cancer data is shown in Fig 11. This gives an overall indicator of whether a given model variant is under- or over-smoothing. If a large portion of the density is greater than or close to 1, this indicates that the model is over-smoothing. Conversely, under-smoothing can be declared for densities close to 0. Note that the values of the relative position of CASIR are capped at -0.2 and 1.2. The distribution of the relative positions for the models fit to the SIDS data is provided as supplementary material (see S15 Fig)

## Model comparison

The GoF criteria are summarised in Figs 4 and 5. These criteria are often used to conduct model selection on the basis of model fit and parsimony. However, as aforementioned, these criteria often don't agree, and can lead to poor model choices. In line with the aims of this paper, we now compare the models based on the GoS statistics using the cut-offs described in Table 1. The full results are provided in the supplementary material, S2 and S3 Tables.

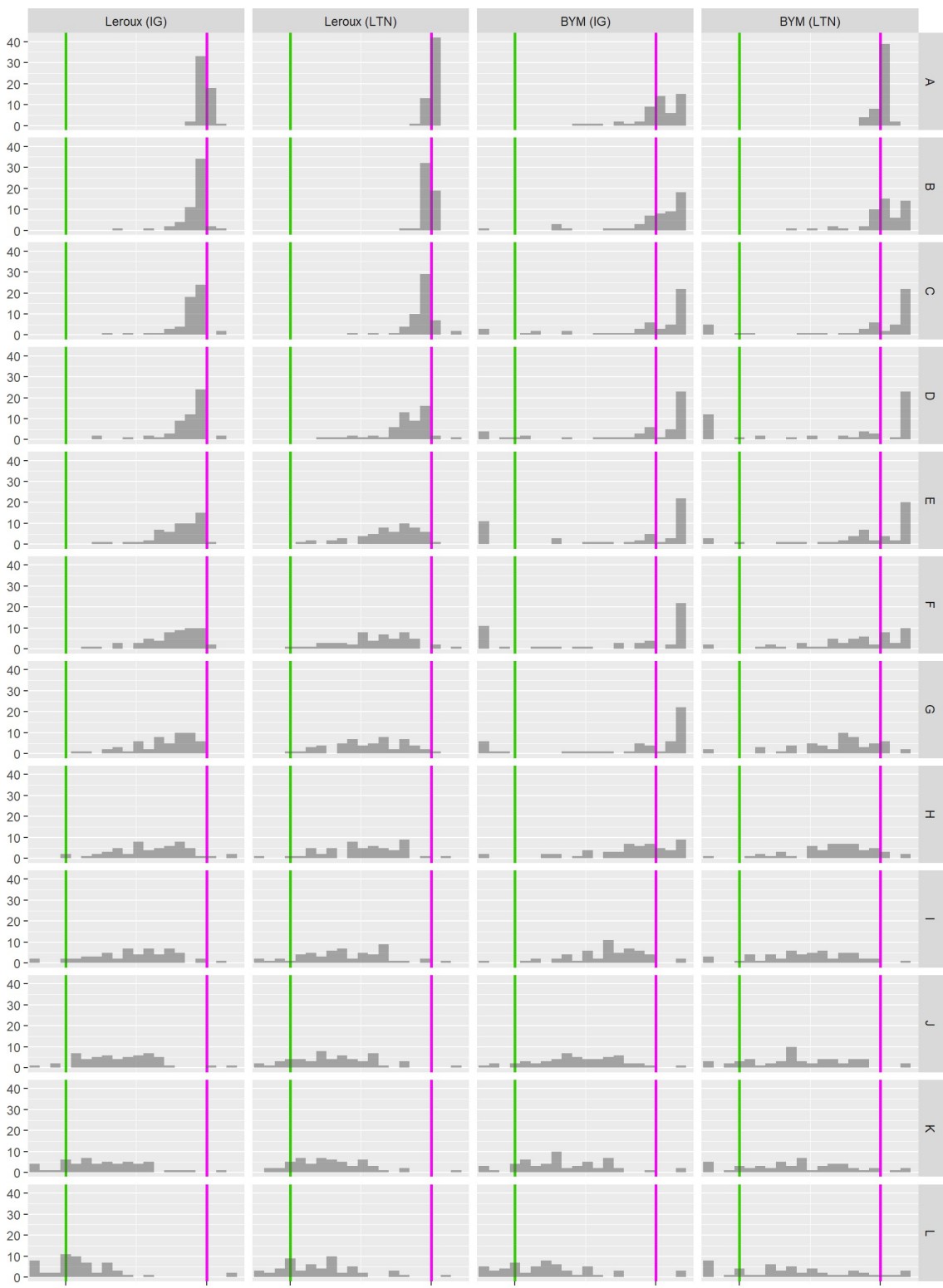

**Fig 11. Distribution of the relative position of the CASIR estimates for each model variant (lip cancer data).** Distributions with substantial density close to 0 indicate under-smoothing; distributions with substantial density close to 1 indicate over-smoothing.

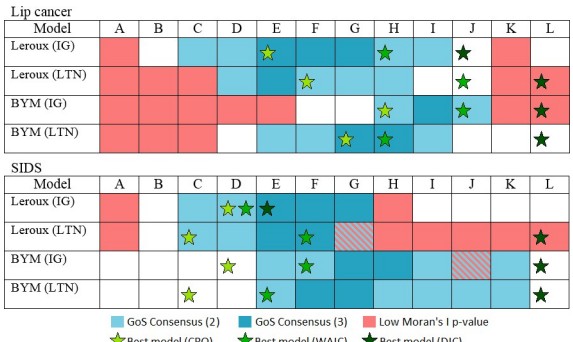

**Fig 12. Consensus of the results based on the three GoS criteria that penalise under-smoothing, and the best models according to the GoF criteria.**

The criteria with unbiased and conservative cut-offs tend to favour under-smoothed models. If more smoothing is desired, then the cut-off that penalises under-smoothing is more appropriate. Despite the differences between these GoS statistics mathematically and differences between the criteria definitions, there is substantial agreement among the results given a particular cut-off (u, c, or pu). Focusing on only the criteria which penalise under-smoothing, Fig 12 provides a consensus result, showing which model variants pass 2 GoS criteria and which pass all 3. The best models under the GoF criteria are included for comparison.

## Discussion

This paper presented three existing GoF measures and proposed five new GoS measures. Each of these measures attempts to quantify one or more important characteristics of a model: goodness of model fit, parsimony, and adequacy of spatial smoothing.

The GoS approaches vary from original proposals to reinventions and modifications of existing ideas. Consequently, there are likely great improvements that can be made, both in defining the statistics and the guidelines for their interpretation. For example, the kurtosis preservation method appeared to be the least reliable GoS measure. This may be improved, for example, if the spatial kurtosis were defined differently. Guidelines for the fraction of spatial variation approach are notably lacking, which may be the main drawback of this otherwise seemingly reliable and relatively simple method.

The third aim of this paper was to compare the results of the GoF and GoS statistics. The criteria used for the GoF statistics were taken from the literature, while the criteria for the GoS criteria were specifically designed to favour models with more smoothing rather than less. Such criteria seem appropriate in practice given the benefits of spatial smoothing. Under these particular criteria, summarised in Fig 12 and presented more fully in Figs 5 and 6 and S2 and S3 Tables, there is a fairly strong consensus among the GoS approaches. Conversely, the GoF criteria rarely agree on the best model, often choosing models with substantially different degrees of smoothing, and even choosing models that are arguably greatly under- or over-smoothed according to the GoS consensus results.

Out of the three GoS approaches forming the consensus, the relative position of the CASIR approach coincided with the consensus (2 or more criteria) 91.7% of the time, the variogram ratio approach coincided 88.2% of the time, and kappa coincided 85.4% of the time. The relative position of CASIR is the only GoS statistic that avoided selecting models with small Moran's I p-values. Thus the relative position of CASIR may be considered the most conservative approach in that other GoS are likely to agree in identifying good models, but not necessarily

vice versa. To achieve the most robust model comparison on the basis of spatial smoothing, it is recommended that multiple GoS methods and even multiple criteria are used. However, the relative position of CASIR is likely to perform well if used independently.

While it is difficult to compare GoF against GoS in the absence of a ground truth, the GoS does appear to identify better models more accurately than GoF based on the consistency of the GoS approaches, and the fact that the model variants were intentionally specified to yield over- and under-smoothed models closest to variants A and L respectively. This is corroborated by visual inference from maps such as those shown in Fig 3.

Using the consensus shown in Fig 12 as the benchmark, the problem with relying on GoF measures to identify the best or even a good model becomes apparent. In particular, DIC tended to identify under-smoothed model variants (variant L identified as best model 6 out of 8 times). The WAIC and CPO criteria tended to align better with the GoS criteria, but still showing a tendency to favour models with less and more smoothing respectively. In fact, the WAIC criteria always choose model variants at least as close to L if not closer than the CPO, and DIC always choose model variants closer to L than the WAIC. Clearly there is a great danger in relying on DIC, and to a lesser extent other GoF measures, to perform model selection among competing spatial models.

While the GoS approaches presented in this paper highlight a very important problem, they offer only simple, empirical solutions to quantifying spatial smoothing. They are by no means model-decision theoretic approaches. However, it is hoped that this demonstration of the challenge will motivate the development of more elaborate solutions, perhaps even combing multiple objective functions into a single utility function to be optimised. In the meantime, these simple GoS approaches should prove useful to researchers evaluating spatial models and performing model selection.

The main limitations of this analysis are the scope of the data, models, and criteria used. Both GoS and GoF criteria require subjective input from the user, usually in the form of cut-offs. While care has been taken to use sensible criteria, different cut-offs may produce different results. Only two datasets and two models were used, albeit with several variants. Another possible extension to this research is to compare these approaches compare across other models and other datasets.

## Supporting information

**S1 Fig. Maps showing the posterior mean estimates of the key model parameters for the Leroux model variants with an IG hyperprior (lip cancer data set, 56 counties of Scotland).** (DOCX)

**S2 Fig. Maps showing the posterior mean estimates of the key model parameters for the Leroux model variants with an LTN hyperprior (lip cancer data set, 56 counties of Scotland).** (DOCX)

**S3 Fig. Maps showing the posterior mean estimates of the key model parameters for the BYM model variants with an IG hyperprior (lip cancer data set, 56 counties of Scotland).** (DOCX)

**S4 Fig. Maps showing the posterior mean estimates of the key model parameters for the BYM model variants with an LTN hyperprior (lip cancer data set, 56 counties of Scotland).** (DOCX)

**S5 Fig. Maps showing the posterior mean estimates of the key model parameters for the Leroux model variants with an IG hyperprior (SIDS data set, 100 counties in North Carolina).**
(DOCX)

**S6 Fig. Maps showing the posterior mean estimates of the key model parameters for the Leroux model variants with an LTN hyperprior (SIDS data set, 100 counties in North Carolina).**
(DOCX)

**S7 Fig. Maps showing the posterior mean estimates of the key model parameters for the BYM model variants with an IG hyperprior (SIDS data set, 100 counties in North Carolina).**
(DOCX)

**S8 Fig. Maps showing the posterior mean estimates of the key model parameters for the BYM model variants with an LTN hyperprior (SIDS data set, 100 counties in North Carolina).**
(DOCX)

**S9 Fig. Model residuals for the lip cancer data set, 56 counties of Scotland.**
(DOCX)

**S10 Fig. Model residuals for the SIDS data set, 100 counties in North Carolina.**
(DOCX)

**S11 Fig. Variograms for each model variant fit to the SIDS data set.** The solid and dashed lines denote the variograms of CASIR and CARSIR respectively, each averaged over the areas. The grey dots denote the area-specific variogram of CASIR. Note that the y-axis has been capped at 1 for clarity.
(DOCX)

**S12 Fig. Kurtosis and roughness for each model variant fit to the SIDS data set.**
(DOCX)

**S13 Fig. Kappa statistic between CASIR and CARSIR for each model variant fit to the SIDS data set, using 3 and 5 discrete categories.** Values close to 0 suggest over-smoothing while values close to 1 suggest under-smoothing.
(DOCX)

**S14 Fig. Fraction of spatial variation for each model variant fit to the SIDS data set.**
(DOCX)

**S15 Fig. Distribution of the relative position of the CASIR estimates for each model variant (SIDS data).** Distributions with substantial density close to 0 indicate under-smoothing; distributions with substantial density close to 1 indicate over-smoothing.
(DOCX)

**S1 Table. The specific values of the hyperparameters $\alpha$, $\eta$, $\nu$, and $\pi$ used to produce the model variants.**
(DOCX)

**S2 Table. Classification of the models based on the GoS criteria (lip cancer data).** A "PASS" indicates that the model variant is neither under- nor over-smoothing under the given criterion (see Table 1). VR = variogram ratio; KP = kurtosis preservation; K = kappa;

RPC = relative position of CASIR; (u) = unbiased; (c) = conservative (less likely to choose under- or over-smoothed models); (pu) = penalise under-smoothing more heavily than over-smoothing.
(DOCX)

**S3 Table. Classification of the models based on the GoS criteria (SIDS data).** A "PASS" indicates that the model variant is neither under- nor over-smoothing under the given criterion (see Table 1). VR = variogram ratio; KP = kurtosis preservation; K = kappa; RPC = relative position of CASIR; (u) = unbiased; (c) = conservative (less likely to choose under- or over-smoothed models); (pu) = penalise under-smoothing more heavily than over-smoothing.
(DOCX)

## Acknowledgments

The authors would like to thank Dr Susanna Cramb for her feedback on earlier drafts of this manuscript.

## Author Contributions

**Conceptualization:** Earl W. Duncan, Kerrie L. Mengersen.

**Data curation:** Earl W. Duncan.

**Formal analysis:** Earl W. Duncan.

**Funding acquisition:** Kerrie L. Mengersen.

**Investigation:** Earl W. Duncan.

**Methodology:** Earl W. Duncan, Kerrie L. Mengersen.

**Project administration:** Kerrie L. Mengersen.

**Resources:** Earl W. Duncan, Kerrie L. Mengersen.

**Software:** Earl W. Duncan.

**Supervision:** Kerrie L. Mengersen.

**Validation:** Earl W. Duncan, Kerrie L. Mengersen.

**Visualization:** Earl W. Duncan.

**Writing – original draft:** Earl W. Duncan.

**Writing – review & editing:** Earl W. Duncan, Kerrie L. Mengersen.

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
