## [Decision Letter · Decision Letter 0]

30 Mar 2020

PONE-D-20-05534

Comparing Bayesian Spatial Models: Goodness-of-smoothing Criteria for Assessing Under- and Over-smoothing

PLOS ONE

Dear Dr Duncan,

Thank you for submitting your manuscript to PLOS ONE. After careful consideration, we feel that it has merit but does not fully meet PLOS ONE’s publication criteria as it currently stands. Therefore, we invite you to submit a revised version of the manuscript that addresses the points raised during the review process.

We would appreciate receiving your revised manuscript by May 14 2020 11:59PM. To enhance the reproducibility of your results, we recommend that if applicable you deposit your laboratory protocols in protocols.io, where a protocol can be assigned its own identifier (DOI) such that it can be cited independently in the future. For instructions see: http://journals.plos.org/plosone/s/submission-guidelines#loc-laboratory-protocols

We look forward to receiving your revised manuscript.

Kind regards,

Qiang Zeng, Ph.D.

Academic Editor

PLOS ONE

Journal Requirements:

2. We note that Figures 2-3 and S1-S10 in your submission contain map images which may be copyrighted. All PLOS content is published under the Creative Commons Attribution License (CC BY 4.0), which means that the manuscript, images, and Supporting Information files will be freely available online, and any third party is permitted to access, download, copy, distribute, and use these materials in any way, even commercially, with proper attribution. For these reasons, we cannot publish previously copyrighted maps or satellite images created using proprietary data, such as Google software (Google Maps, Street View, and Earth). For more information, see our copyright guidelines: http://journals.plos.org/plosone/s/licenses-and-copyright.

1.    You may seek permission from the original copyright holder of Figures 2-3 and S1-S10 to publish the content specifically under the CC BY 4.0 license. 

Reviewers' comments:

Reviewer's Responses to Questions

**Comments to the Author**

1. Is the manuscript technically sound, and do the data support the conclusions?

Reviewer #1: Yes

Reviewer #2: Yes

2. Has the statistical analysis been performed appropriately and rigorously? 

Reviewer #1: Yes

Reviewer #2: Yes

3. Have the authors made all data underlying the findings in their manuscript fully available?

Reviewer #1: Yes

Reviewer #2: Yes

4. Is the manuscript presented in an intelligible fashion and written in standard English?

Reviewer #1: Yes

Reviewer #2: Yes

5. Review Comments to the Author

Reviewer #1: The paper proposed several goodness-of-smoothing criteria for assessing the bayesian spatial model, and compared them against traditional goodness-of-fit statistics on real data. The topic is very important and promising. The study is appropriate for publication in the journal of Plos One with only minor revisions.

As a researcher of traffic statistical analysis, I fully understand that Bayesian spatial models are widely used in accident statistical modelling (some recent references are listed below). It is suggested that latest development and applications for Bayesian spatial models (such as used in traffic fields) can be briefly introduced.

Apostolos Z, George Y. A review of spatial approaches in road safety. Accident Analysis and Prevention. https://doi.org/10.1016/j.aap.2019.105323.

Zeng Q, Gu W , Zhang X, et al. Analyzing freeway crash severity using a Bayesian spatial generalized ordered logit model with conditional autoregressive priors. Accident Analysis and Prevention, 2019, 127, 87-95.

Zeng Q, Wen H, Huang H, et al. A Bayesian spatial random parameters Tobit model for analyzing crash rates on roadway segments[J]. Accident Analysis and Prevention, 2017, 100: 37-43.

Ma Q, Yang H Xie K, et al. Taxicab crashes modeling with informative spatial autocorrelation. Accident Analysis and Prevention, 2019, 131, 297-307.

The abstract is suggested to rewritten with four parts: objective, methods, results and conclusions.

Reviewer #2: This study proposed several methods for quantifying degree of smoothing. By comparing these methods against commonly used goodness-of-fit measures, the authors demonstrated the inadequacy of depending solely on goodness-of-fit criteria for spatial model selection. The topic is interesting and worthy of investigation. The whole manuscript is well structured and easy to follow. Before suggesting it publication, several issues, however, need to be well addressed.

1. The commonly used inverse-Gamma (α,η) prior for variance parameter is sensitive to the values of α and η if the true variance is close to zero (Gelman, 2006). In addition to specifying different combination of values of α and η to fit models with varying degrees of smoothing, the authors are therefore suggested to use a uniform (0, M) prior for σ_s as a benchmark.

2. The definition of spatial correlation should have an effect on the degree of smoothing. Despite simplification and easy to manipulate, use of binary first-order adjacency weights that only areas with common borders are assumed to be spatially correlated is indeed a strong assumption, especially in empirical case studies without validation. This limitation should be highlighted at the end of manuscript.

Reference

Gelman, A., 2006. Prior distributions for variance parameters in hierarchical models. Bayesian Anal. 1, 515-533.

6. PLOS authors have the option to publish the peer review history of their article (what does this mean?). If published, this will include your full peer review and any attached files.

Reviewer #1: No

Reviewer #2: No

---

## [Author Response · Author response to Decision Letter 0]

25 Apr 2020

Reviewer 1

Comment 1: As a researcher of traffic statistical analysis, I fully understand that Bayesian spatial models are widely used in accident statistical modelling (some recent references are listed below). It is suggested that latest development and applications for Bayesian spatial models (such as used in traffic fields) can be briefly introduced.

Apostolos Z, George Y. A review of spatial approaches in road safety. Accident Analysis and Prevention. https://doi.org/10.1016/j.aap.2019.105323.

Zeng Q, Gu W , Zhang X, et al. Analyzing freeway crash severity using a Bayesian spatial generalized ordered logit model with conditional autoregressive priors. Accident Analysis and Prevention, 2019, 127, 87-95.

Zeng Q, Wen H, Huang H, et al. A Bayesian spatial random parameters Tobit model for analyzing crash rates on roadway segments[J]. Accident Analysis and Prevention, 2017, 100: 37-43.

Ma Q, Yang H Xie K, et al. Taxicab crashes modeling with informative spatial autocorrelation. Accident Analysis and Prevention, 2019, 131, 297-307.

Response 1: The authors thank the reviewer for these useful references. We found the first and third reference particularly relevant for the motivation of this research and have added these to the introduction as suggested, on line 81.

Comment 2: The abstract is suggested to rewritten with four parts: objective, methods, results and conclusions.

Response 2: The abstract has been rewritten in four parts as suggested. The new abstract now reads:

Background:

Many methods of spatial smoothing have been developed, for both point data as well as areal data. In Bayesian spatial models, this is achieved by purposefully designed prior(s) or smoothing functions which smooth estimates towards a local or global mean. Smoothing is important for several reasons, not least of all because it increases predictive robustness and reduces uncertainty of the estimates. Despite the benefits of smoothing, this attribute is all but ignored when it comes to model selection. Traditional goodness-of-fit measures focus on model fit and model parsimony, but neglect “goodness-of-smoothing”, and are therefore not necessarily good indicators of model performance. Comparing spatial models while taking into account the degree of spatial smoothing is not straightforward because smoothing and model fit can be viewed as opposing goals. Over- and under-smoothing of spatial data are genuine concerns, but have received very little attention in the literature. 

Methods:

This paper aims to demonstrates the problem with spatial model selection based solely on goodness-of-fit, to propose by proposing several methods for quantifying the degree of smoothing, and to compare these methods. Several commonly used spatial models are fit to real data, and subsequently compared using the against goodness-of-fit and goodness-of-smoothing statistics on real data. 

Results:

The proposed goodness-of-smoothing statistics show substantial agreement in the task of model selection, and tend to avoid models that over- or under-smooth. Conversely, the traditional goodness-of-fit criteria often don’t agree, and can lead to poor model choice. In particular, the well-known deviance information criterion tended to select under-smoothed models.

Conclusions:

Some of the goodness-of-smoothing methods may be improved with modifications and better guidelines for their interpretation. However, these proposed goodness-of-smoothing methods offer researchers a solution to spatial model selection which is easy to implement. Moreover, they highlight the danger in relying on goodness-of-fit measures when comparing spatial models.

Reviewer 2

Comment 1: The commonly used inverse-Gamma (α,η) prior for variance parameter is sensitive to the values of α and η if the true variance is close to zero (Gelman, 2006). In addition to specifying different combination of values of α and η to fit models with varying degrees of smoothing, the authors are therefore suggested to use a uniform (0, M) prior for σ_s as a benchmark.

Response 1: The authors implemented model variants using a uniform (0, M) prior as suggested. Specifically, the BYM and Leroux models were re-run on both data sets, each using variants of the uniform prior by changing the value of M, ranging from 0.5 to 10^4 (larger values resulted in priors too vague for the sampler to produce samples from). The results showed very little variation between model variants. This seems to be consistent with Gelman (2006), namely that for a finite but sufficiently large M, inferences are not sensitive to this choice of M. Unfortunately, there is no guarantee that such models will produce a model with an adequate degree of smoothing. In the case of the Scottish lip cancer, the models with the uniform prior were actually quite good in terms of the goodness-of-smoothing compared to the best inverse-gamma and left-truncated normal priors. This was reflected by the GoS criteria as well as a visual inspection of the risk surface and associated maps. Conversely, for the North Carolina SIDS data, the models with the uniform prior resulted in slight over-smoothing for the Leroux variants (comparable to variant D of the Leroux IG and LTN models), and severe over-smoothing for the BYM variants (comparable to variants C of the BYM IG and LTN models). It is comforting to see the proposed GoS criteria performing as expected for the uniform model variants. However, due to the inability to produce variation in the results because of the insensitivity to the hyper –parameter, we have not included these results in the paper. The authors do agree with the reviewer that trying other model specifications would be useful, but this has already been noted as a limitation and a recommendation for future research (lines 805-806 of the revised manuscript).

Comment 2: The definition of spatial correlation should have an effect on the degree of smoothing. Despite simplification and easy to manipulate, use of binary first-order adjacency weights that only areas with common borders are assumed to be spatially correlated is indeed a strong assumption, especially in empirical case studies without validation. This limitation should be highlighted at the end of manuscript. 

Response 2: The authors acknowledge that this is a strong assumption. The authors have stated this assumption up front in the introduction as one of the three constraints imposed on this study. Given the focus of the study is on proposing and comparing the goodness-of-smoothing criteria, we do not consider it a limitation of this study per se, but rather that consideration of other spatial weights specifications are out of scope. Consequently, no changes have been made to the manuscript.

---

## [Editor Report · Decision Letter 1]

28 Apr 2020

Comparing Bayesian Spatial Models: Goodness-of-smoothing Criteria for Assessing Under- and Over-smoothing

PONE-D-20-05534R1

Dear Dr. Duncan,

We are pleased to inform you that your manuscript has been judged scientifically suitable for publication and will be formally accepted for publication once it complies with all outstanding technical requirements.

With kind regards,

Qiang Zeng, Ph.D.

Academic Editor

PLOS ONE
---

## [Editor Report · Acceptance letter]

7 May 2020

PONE-D-20-05534R1 

Comparing Bayesian Spatial Models: Goodness-of-smoothing Criteria for Assessing Under- and Over-smoothing 

Dear Dr. Duncan:

I am pleased to inform you that your manuscript has been deemed suitable for publication in PLOS ONE. Congratulations! Your manuscript is now with our production department. 

With kind regards,

on behalf of

Dr. Qiang Zeng 

Academic Editor

PLOS ONE